# Solution structure of mouse HBS1L/SKI7-specific UBA domain in complex with ubiquitin: Implications for stalled ribosome recognition

Nobukazu Nameki[1]☯, Fahu He[2]☯, Minako Okada[3], Mari Takahashi[2,4,5,6], Kengo Tsuda[2,4,5], Takashi Nagata [4,7], Peter Güntert[8,9,10,11], Naohiro Kobayashi[2,6,12], Takanori Kigawa[2,5,6], Mikako Shirouzu[2,4,5,6], Akiko Tanaka [2,4,5], Shigeyuki Yokoyama [2¤], Yutaka Muto[2,3,5]*, Kanako Kuwasako [2,3,5]*

1 Division of Molecular Science, Graduate School of Science and Technology, Gunma University, Kiryu, Gunma, Japan, 2 RIKEN, Systems and Structural Biology Center, Turumi-ku, Yokohama, Japan, 3 Faculty of Pharmacy and Research Institute of Pharmaceutical Sciences, Musashino University, Nishitokyo, Tokyo, Japan, 4 RIKEN Center for Life Science and Technologies, Tsurumi-ku, Yokohama, Japan, 5 RIKEN Center for Biosystems Dynamics Research, Tsurumi-ku, Yokohama, Japan, 6 Present address: RIKEN Center for Integrative Medical Sciences, Tsurumi-ku, Yokohama, Japan, 7 Institute of Advanced Energy and Graduate School of Energy Science, Kyoto University, Uji, Kyoto, Japan, 8 Tatsuo Miyazawa Memorial Program, RIKEN Genomic Sciences Center, Yokohama, Japan, 9 Institute of Biophysical Chemistry, Goethe-University Frankfurt am Main, Frankfurt am Main, Germany, 10 Institute of Molecular Physical Science, ETH Zurich, Zurich, Switzerland, 11 Department of Chemistry, Tokyo Metropolitan University, Hachioji, Tokyo, Japan, 12 RIKEN Yokohama NMR Facility, 1-7-22 Tsurumi-ku, Yokohama, Japan

☯ These authors contributed equally to this work.
¤ Current address: Department of Structural Biology and Biochemistry, Institute of New Industry Incubation, Institute of Science Tokyo, Bunkyo-ku, Tokyo, Japan; Department of Drug Target Protein Research, Shinshu University School of Medicine, Matsumoto, Nagano, Japan
* k063012@ptf.musashino-u.ac.jp (YM); kanameki@musashino-u.ac.jp (KK)

## Abstract

Human HBS1L and SKI7 (HBS1LV3) are isoforms encoded by the same gene. HBS1L forms a complex with PELO to recognize ribosomes stalled on non-stop mRNAs and promotes ribosome splitting, whereas SKI7 acts as a bridge between the exosome and the SKI complex to mediate mRNA decay on stalled ribosomes. Despite substantial differences in the sequence and function of their C-terminal regions, the two isoforms share an identical N-terminal domain (termed UBAh) that resembles the ubiquitin binding UBA and CUE domains (collectively referred to as the three-helix bundle ubiquitin-binding [THB–Ub] group). Although UBAh has been predicted to interact with ubiquitin moieties attached to the small subunits of stalled ribosomes, evidence for its interaction with ubiquitin is lacking. Herein, we report the NMR structure of the mouse UBAh–ubiquitin complex. UBAh adopts a three-helix bundle architecture (α1–α2–α3) with unique connecting loops. The hydrophobic patch in UBAh interacts with the Ile44-centered hydrophobic patch of ubiquitin in a binding mode nearly identical to that of the UBA and CUE domains. In contrast, the α1/

**Data availability statement:** All relevant data are within the manuscript and its Supporting Information files. The minimal dataset is available in the Biological Magnetic Resonance Data Bank (BMRB) (https://bmrb.io/) under accession numbers 36786 and 36787, and in the Protein Data Bank (PDB) (https://www.rcsb.org/) under accession codes 9WPR and 9WPS.

**Funding:** This work was supported by JSPS KAKENHI (grant number 22K06105, to K.K.) and a Daigaku Tokubetsu Kenkyuhi grant from Musashino University (to Y.M. and K.K.). The funders had no role in study design, data collection and analysis, decision to publish, or preparation of the manuscript.

**Competing interests:** The authors declare no conflict of interests.

α2 loop contains a distinctive double β-turn that accommodates the protrusion of the ubiquitin β-turn. The hallmark motif of UBAh, located within and downstream of this loop, was identified as VLGD/E. HSQC titration experiments yielded a dissociation constant of approximately 50 μM for ubiquitin. These findings demonstrate that UBAh specifically interacts with ubiquitin in vitro, providing structural insights into its potential role in recruiting HBS1L–PELO and SKI7 to stalled ribosomes.

## Introduction

Ribosomes stall on mRNAs for various reasons, particularly under stressful conditions. Stalled ribosomes disrupt translational homeostasis and impair cellular growth and function. To counteract such disturbances, cells have evolved a robust translational quality control network that monitors aberrant translation events and ensures the proper resolution of defective ribosome complexes (for recent comprehensive reviews, see [1–4]). This system comprises multiple interconnected pathways, including non-stop decay (NSD), no-go decay (NGD), ribosome rescue, and ribosome-associated quality control (RQC) (Fig 1).

In the present study, we focused on the N-terminal domain of HBS1L. In mammals, HBS1L (682 residues in mice) has an alternatively spliced isoform, SKI7 (also known as HBS1LV3), which shares an identical N-terminal domain, but differs in the remaining region, leading to distinct functions. Here, we describe HBS1L; SKI7 will be addressed in detail in the Discussion section. HBS1L cooperates with PELO and GTP to form the HBS1L–PELO complex (Hbs1–Dom34 in the budding yeast *Saccharomyces cerevisiae*) [5–8]. Its fundamental function is to enter the vacant A site of ribosomes stalled on mRNAs lacking a stop codon (non-stop mRNAs) and promote their splitting into small (40S) and large (60S) ribosomal subunits (Fig 1). PELO senses the absence of mRNA at site A, whereas HBS1L hydrolyzes GTP and induces conformational changes. Subsequently, ABCE1 (Rli1 in yeast), an ATP-binding cassette (ABC)-type ATPase, facilitates the dissociation of the stalled 80S ribosome into 40S and 60S subunits [9–12].

The HBS1L–PELO complex appears to be involved in both the NSD and NGD pathways (Fig 1). The NSD pathway is initiated by the translation of non-stop mRNAs, which often arise from premature polyadenylation or the aberrant cleavage of mRNAs [13,14]. Ribosomes that translate non-stop mRNAs consequently stall at the 3′ end, frequently within poly(A) tails [15]. The resulting non-stop ribosomes are well-established targets of the HBS1L–PELO complex, which promotes their splitting. The aberrant mRNAs that are bound to these stalled ribosomes are primarily and rapidly degraded by the cytoplasmic exosome (3′→5′) and the SKI complex [8,15,16]. The split 60S subunit is thought to be processed via the RQC pathway, which involves recognition of the 60S subunit carrying a peptidyl-tRNA, K48-linked ubiquitination of the nascent polypeptide, cleavage of peptidyl-tRNA, extraction of the ubiquitinated polypeptide, and subsequent proteasome-mediated degradation [3,4,17]. In contrast, the NGD pathway is triggered by collisions between trailing

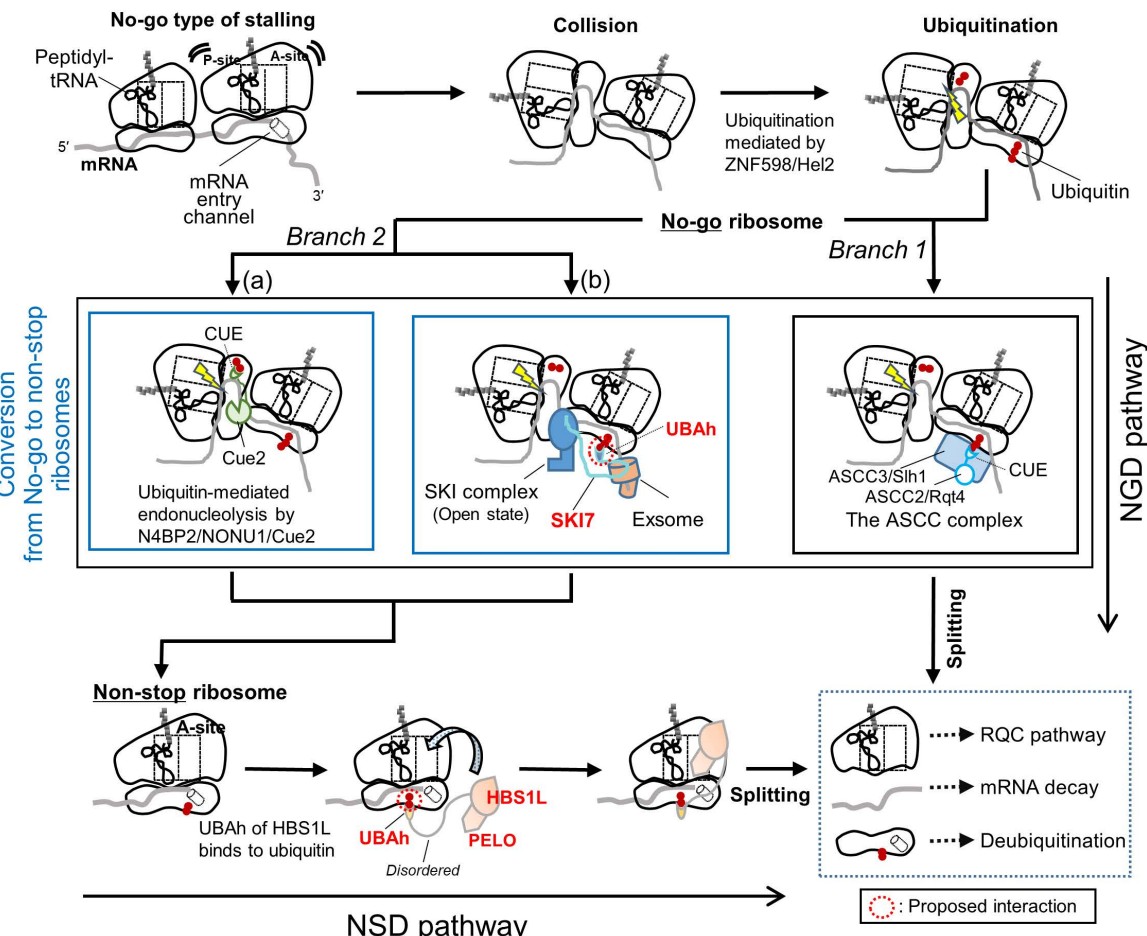

**Fig 1. Proposed model for the roles of the UBAh domains of HBS1L and SKI7 (HBS1LV3) in ubiquitin-mediated stalled ribosome rescue.** When a trailing ribosome collides with a stalled one, a characteristic dimer structure (disome) often forms. This structure is recognized by the ubiquitin ligase ZNF598/Hel2, which ubiquitinates the small subunit of multiple ribosomal proteins, with uS10 (RPS20) as the principal target. Subsequently, the NGD pathway is initiated and diverges into mRNA cleavage-dependent and -independent branches (1 and 2, respectively), although this is likely to be more complex in vivo. Branch 1 is further subdivided into two sub-branches (1a and 1b). In branch 1a, the endonuclease N4BP2/NONU-1/Cue2 typically cleaves mRNA between colliding ribosomes, thereby converting the trailing ribosome into a non-stop ribosome. In yeast, the two N-terminal CUE domains have been reported to recognize polyubiquitination of eS7 (simplified in the figure). In branch 1b, the SKI complex recognizes the disome and recruits the RNA exosome via the bridging protein SKI7, promoting 3′-to-5′ degradation of the mRNA. In branch 2, the ribosome-splitting factor ASCC complex (RQT in yeast) binds to the disome via an interaction between the CUE domain of ASCC2/Rqt4 and ubiquitin. In this case, non-stop ribosomes are unlikely to be generated, thereby activating the RQC pathway. Branch 1 ultimately results in the formation of non-stop ribosomes, which are likely ubiquitinated on the small subunit. Ubiquitin may be recognized by the UBAh domain of HBS1L in complex with PELO, which facilitates the recruitment of the functional core of the complex to the A site of non-stop ribosomes and promotes their efficient splitting. In addition, ubiquitin may be recognized by the UBAh domain of SKI7, which enhances the recognition of aberrant ribosomes and strengthens the interaction between the exosome and the 40S subunit. Note that, for simplicity, the pathway in which non-stop mRNAs are intrinsically generated and translated by ribosomes, leading to the formation of non-stop ribosomes, is omitted.

and stalled ribosomes (no-go ribosomes), typically caused by mRNA obstacles, such as stable secondary structures or damaged bases [6,18] (Fig 1). These collisions generate a stable di-ribosome structure or disome, which is recognized by the E3 ubiquitin ligase ZNF598 (Hel2 in yeast) via a rotated interface formed between the two 40S subunits [19–22]. Upon collision, several 40S proteins near the mRNA entry site are ubiquitinated in yeast, *Caenorhabditis elegans*, and humans [23]. The principal target of the E3 ubiquitin ligase ZNF598/Hel2 is uS10 (RPS20), which undergoes K63-linked

polyubiquitination; other ubiquitination sites vary among species (S1A and S1B Fig). Subsequently, this pathway diverges into two branches, namely the mRNA cleavage–independent and cleavage-dependent pathways, the latter of which is further subdivided into two branches (Fig 1). In the former branch, the ASCC complex (RQT in yeast) recognizes these ubiquitin modifications and splits the collided ribosomes [24–26]. In one of the latter branches (the other is described in the Discussion section), in yeast, the endonuclease Cue2 cleaves mRNA near the A-site or upstream of the collided ribosome, depending on whether uS10 (RQC-coupled) or eS7 (RQC-uncoupled) is polyubiquitinated [27]. In the nematode *C. elegans*, the Cue2 ortholog NONU-1 (human N4BP2) similarly cleaves mRNA following ZNF598-dependent ubiquitination, requiring its CUE domain for binding to specific ribosomal proteins [28]. This cleavage generates secondary non-stop ribosomes, which may be resolved by the HBS1L–PELO complex. Importantly, it is plausible that these non-stop ribosomes undergo ubiquitination. As the NSD and NGD pathways are increasingly regarded as being more closely interconnected than previously anticipated [3,4], the HBS1L–PELO complex may play a more prominent role in the translational quality control network.

HBS1L is divided into two structural domains connected by a disordered region (Fig 2A; S1A Fig). The C-terminal region contains a translational eukaryotic elongation factor 1A (eEF1A)-like GTPase domain and a PELO-binding region (hereafter referred to as the GTPase domain). The N-terminal region begins with stretches of acidic residues (as described in the Discussion), followed by a three-helix bundle that resembles the helical arrangement observed in ubiquitin-associated (UBA) domains and coupling of ubiquitin conjugation to ER degradation (CUE) domains [30]. In this study, we classified the UBA and CUE domains into a single group of the ubiquitin-binding domains based on their shared three-helix bundle architecture and designated this group as the three-helix bundle ubiquitin-binding (THB–Ub) group. These domains are also found in several proteins involved in the translational quality control system, including Cue2 (which contains three CUE domains and one UBA domain arranged in the order CUE–CUE–UBA–CUE) and ASCC2 (which contains a single CUE domain in its middle region) [27,31–33]. These domains specifically recognize and bind to the ubiquitin moieties of the K63-linked ubiquitin chains on the 40S subunit. In this study, the three-helix bundle in the N-terminus of HBS1L is referred to as a ubiquitin-associated domain specific to HBS1L, termed "UBAh" (Fig 2A). Based on the significant structural similarities between the UBA, CUE, and UBAh domains, it has been hypothesized that UBAh binds to ubiquitin, likely in the form of polyubiquitin, on its 40S subunit [28]. If such ubiquitin binding occurs, it facilitates the recruitment of the functional core of the HBS1L–PELO complex to the A site of a non-stop ribosome.

However, the information currently available on the UBAh domain is limited. Two cryo-electron microscopy (cryo-EM) analyses of stalled ribosomes in the absence of ubiquitin in *S. cerevisiae* revealed that the C-terminal GTPase domain of HBS1L is positioned at the A-site in a manner similar to that of eRF3 [7,34] (S1A and S1C Fig). UBAh has been reported to bind to a cavity near known ubiquitination sites (uS10 and uS3) on the 40S subunit, seemingly functioning as an anchor. In contrast, in *C. elegans*, the deletion of the corresponding UBAh region in *hbs-1*, along with much of the linker region, has been shown to result in no discernible phenotype [28]. These findings indicated that the N-terminal region of HBS-1 is dispensable for NGD in *C. elegans*. Thus, the findings in *S. cerevisiae* and *C. elegans* provide no evidence of an interaction between UBAh and ubiquitin. Nevertheless, its structural resemblance to established ubiquitin-binding domains opens up the possibility that UBAh recognizes ubiquitination in other organisms, particularly metazoans.

In the present study, we provide solution structures for the mouse HBS1L UBAh–ubiquitin complex and UBAh alone, as determined by nuclear magnetic resonance (NMR) spectroscopy. Structure-based sequence alignments of UBAh were performed across HBS1L orthologs from fungi and animals. The structure of the complex showed that the UBAh domain specifically interacts with ubiquitin via the Ile44-centered hydrophobic patch, a binding mode nearly identical to that of the UBA and CUE domains. However, marked sequence and structural variation relative to these two domains was observed in the α1/α2 loop, where we identified a key motif that distinguishes UBAh from UBA and CUE domains. Taken together, these findings establish UBAh as a novel member of the THB–Ub group. Notably, two exceptional species appear to exist: *S. cerevisiae*, in which UBAh fails to bind ubiquitin owing to structural differences, and *C. elegans,* in which HBS1L lacks

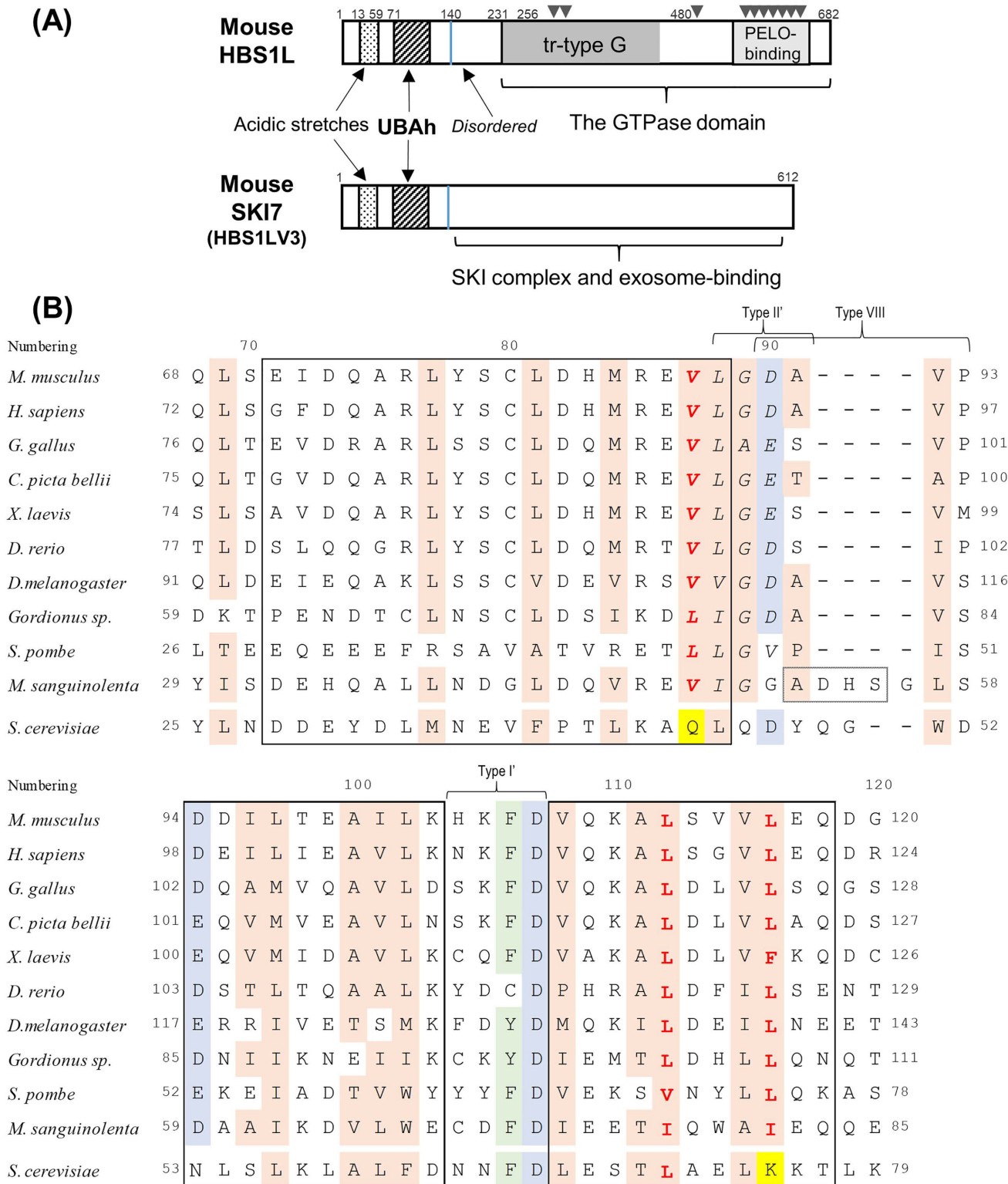

**Fig 2. Domain architecture of mouse HBS1L and SKI7 and structure-based sequence alignment of UBAh domains from various eukaryotes.**
(A) Schematic representation of the domain architecture of mouse HBS1L and SKI7. The N-terminal region (residues 1–140, indicated by a blue line)

is identical in HBS1L and SKI7, and contains acidic stretches as well as the UBAh domain. The C-terminal region is composed of a translational-type guanine nucleotide-binding domain (tr-type G) and a region responsible for PELO binding. For simplicity, in this study it is referred to as the GTPase domain. The UBAh and GTPase domains are connected by a disordered region. SKI7 contains the UBAh domain and a distinct C-terminal region that mediates interactions with both the exosome and the SKI complex. (B) Amino acid sequence alignment of the UBAh domain region of HBS1L/SKI7 from various eukaryotes. Alignment was performed using standard alignment software and then manually modified based on the structure of the mouse UBAh domain. *C. elegans* HBS1L was omitted because its N-terminal region could not be aligned owing to substantial sequence divergence. Residue numbering above the alignment is based on mouse HBS1L. Highly conserved residues across species are highlighted: hydrophobic residues in light beige (Thr is also included in this study because its side chain contains a methyl group); Asp or Glu in pale blue; and Phe or Tyr in pale green. Residues shown in bold red are essential for interaction with ubiquitin. Among the interaction sites, residues in *S. cerevisiae* that differ from those of other species are highlighted in yellow. Residues in α-helices are shown in black boxes. Residues predicted by the AlphaFold program [29] to form an additional α-helix in the α1/α2 loop of *M. sanguinolenta* are boxed with dotted lines. Residues of the UBAh hallmark motif are shown in italics. Accession codes are provided in Materials and Methods.

the UBAh domain, as indicated by sequence analysis. Therefore, with a few apparent exceptions, UBAh represents a broadly conserved ubiquitin-binding domain of HBS1L in fungi and animals.

## Materials and methods

### Accession codes

The accession codes of HBS1L proteins for sequence alignments, obtained from UniProt and/or NCBI, were as follows: *Mus musculus* (Q69ZS7); *Homo sapiens* (Q9Y450); *Gallus gallus* (E1BW59); *Chrysemys picta bellii* (A0A8C3HXA3); *Xenopus laevis* (Q6GNS6); *Danio rerio* (F1Q5Z6); *Drosophila melanogaster* (Q9W074); *Gordionus sp.* (XP_065321722.1); *Schizosaccharomyces pombe* (O74774); *Mycena sanguinolenta* (KAF7355041.1); *S. cerevisiae* (P32769); *C. elegans* (P90922/NP_001021556.2).

### Sample preparation

DNA encoding residues 51–120 of mouse HBS1L (*M. musculus*; Q69ZS7) was subcloned from a full-length mouse cDNA clone by polymerase chain reaction. The DNA fragment was cloned into the expression vector pCR2.1 (Invitrogen) to produce a fusion protein containing an N-terminal His affinity tag, a tobacco etch virus (TEV) protease cleavage site, and a GSSGSSG linker sequence preceding the HBS1L sequence. For ubiquitin, the encoded DNA was cloned into the pCR2.1 vector and expressed as a fusion protein with the same N-terminal sequence. Consequently, the expressed protein containing UBAh used in this study included the artificial tag-derived sequences GSSGSSG at the N-terminus and SGPSSG at the C-terminus, whereas the ubiquitin protein contained only the GSSGSSG sequence at its N-terminus.

$^{13}$C/$^{15}$N-labeled fusion proteins were synthesized using a cell-free protein expression system [35,36]. The lysate was clarified by centrifugation at $16,000 \times g$ for 20 min and filtered through a 0.45-µm membrane (Merck Millipore). The clarified lysate was applied to a His-Trap column (Cytiva) and eluted using a 12–500 mM imidazole gradient. The affinity tag was then removed by incubation with TEV protease for 1 h at 30°C. The tag-free proteins were further purified using Superdex 75 gel filtration chromatography (Cytiva). The purified HBS1L and ubiquitin samples were each concentrated to approximately 1.0 mM in 20 mM sodium phosphate buffer (pH 6.0), containing 100 mM NaCl, 1 mM 1,4-DL-dithiothreitol-d$_{10}$, and 0.02% NaN$_3$ (in 90% H$_2$O/10% D$_2$O) using an Amicon Ultra-15 filter (3,000 MWCO; Merck Millipore). For NMR measurements of the complex structure, HBS1L and ubiquitin solutions were mixed at a molar ratio of 1:1 and concentrated to approximately 1.0 mM both for each protein.

### NMR spectroscopy and structure calculations

All NMR data were acquired at 25°C on Bruker 600 MHz and Bruker 800 MHz spectrometers and processed using NMRPipe [37]. Processed data were analyzed using NMRView [38] and KUJIRA [39]. Two-dimensional (2D) $^1$H–$^{13}$C and

$^1$H–$^{15}$N HSQC spectra, as well as three-dimensional (3D) HNCO, HN(CA)CO, HNCA, HN(CO)CA, HNCACB, CBCA(CO)NH, HBHA(CO)NH, H(CCCO)NH, (H)CC(CO)NH, HCCH-TOCSY, HCCH-COSY, and CCH-TOCSY spectra acquired on a Bruker 600 MHz spectrometer were used to assign all carbon, nitrogen, and hydrogen atoms of the $^{13}$C/$^{15}$N-labeled samples [40,41].

Furthermore, 2D $^1$H–$^{13}$C and $^1$H–$^{15}$N HSQC spectra, and 3D $^{15}$N- and $^{13}$C-edited NOESY spectra acquired on a Bruker 800 MHz spectrometer were used for structural calculations. The proton shifts were indirectly referenced using the $^1$H resonance frequency of the methyl group of DSS (the chemical shift reference of $^1$H was based on the proton of water (4.821 ppm at 298 K)), and the $^{13}$C and $^{15}$N resonances were indirectly calibrated based on their gyromagnetic ratios, according to IUPAC recommendations [42]. The nuclear Overhauser effect (NOE) peaks from the $^{15}$N- and $^{13}$C-edited NOESY spectra with a mixing time of 80 ms were converted into distance constraints for structure calculations. The 3D structure of the protein complex was determined by combining automated NOESY cross-peak assignment and structural calculations with torsion angle dynamics [43] implemented in CYANA 2.1 [44]. The dihedral angle restraints for ϕ and ψ were obtained from the main-chain and $^{13}$C$^\beta$ chemical shift values using the TALOS program [45] and by analyzing the NOESY spectra. Stereospecific assignments for the isopropyl methyl and methylene groups were determined based on the patterns of the inter- and intra-residual NOE intensities [46]. Hydrogen-bond restraints were not used in this study. The structure calculation started with 400 randomized conformers using a standard CYANA-simulated annealing schedule with 40,000 dynamic torsion angle steps per conformer [47]. Further refinements by restrained molecular dynamics simulations, followed by restrained energy minimization, were performed for the 80 conformers with the lowest final CYANA target function values using the Amber24 program with an ff19SB force field and generalized Born model [48], as previously described [49]. Finally, 20 conformers with the lowest Amber energy values were selected as the final structures for deposition in the Worldwide Protein Data Bank. PROCHECK-NMR [50] and MOLMOL [51] were used to validate and visualize the deposited structures.

Images of ribosome-containing structures were generated using PyMOL (PyMOL Molecular Graphics System, version 2.0, Schrödinger, LLC). Other structures were generated using MOLMOL, unless otherwise stated.

The atomic coordinates for the ensemble of 20 energy-refined NMR conformers representing the solution structures of free UBAh and the UBAh–ubiquitin complex were deposited in the Worldwide Protein Data Bank under the accession codes 9WPR and 9WPS, respectively. Chemical shift assignments for free UBAh and the UBAh–ubiquitin complex were deposited in the BMRB database under accession numbers 36786 and 36787, respectively.

### Chemical shift perturbation experiments

$^{15}$N-labeled UBAh (0.1 mM) was dissolved in NMR buffer as described above, and unlabeled ubiquitin was added stepwise to a final molar ratio of 1:3. At each step, a 2D $^1$H-$^{15}$N HSQC spectrum was acquired. The weighted chemical shift change in the amide $^1$H and $^{15}$N for each residue, $\Delta\delta(^{15}N + ^1H_N)$, was calculated as $[(\Delta\delta_{15N}/6.5)^2 + \Delta\delta_{1H}^2]^{1/2}$, where $\Delta\delta_{15N}$ and $\Delta\delta_{1H}$ represent the observed chemical shift changes of the amide nitrogen and proton, respectively. The dissociation constant ($K_d$) for UBAh–ubiquitin binding was determined by fitting the chemical shift perturbation data to the following equation [52]:

$$\Delta\delta_{obs} = \Delta\delta_{max} \{[K_d + P_0(1 + x)] - [(K_d + P_0(x + 1))^2 - 4P_0^2 x]^{1/2}\}/(2P_0),$$

where $\Delta\delta_{obs}$ is the weighted chemical shift change from the free state; $x$ is the molar ratio of UBAh to ubiquitin; $P_0$ is the initial concentration of the labeled protein (UBAh); and $\Delta\delta_{max}$ is the maximum chemical shift change. Global nonlinear regression analysis, assuming a single binding-site model, was performed using GraphPad Prism 6 based on titration data from three residues.

## Results

### Determination of the solution structure of the UBAh–ubiquitin complex

The structure of UBAh was determined using residues 51–120 of mouse HBS1L, whereas that of ubiquitin was determined using its full-length form (79 residues) with an N-terminal seven-residue tag. Mouse UBAh and ubiquitin share 84% and 100% sequence identity with their human counterparts, respectively. Each protein was overexpressed in *Escherichia coli* and labeled with $^{15}$N and/or $^{13}$C or left unlabeled, depending on the type of experiment.

The solution structure of the UBAh–ubiquitin complex was determined at a 1:1 molar ratio of the two proteins using standard multidimensional heteronuclear NMR spectroscopy. The completeness of the backbone and side-chain resonance assignments is summarized in S1 Table. The assigned 2D $^{1}$H–$^{15}$N HSQC spectrum of the tagged UBAh protein in the presence of ubiquitin is shown in S2 Fig. Tertiary structures were calculated using the CYANA software package [44] based on 3,558 $^{1}$H–$^{1}$H distance restraints derived from NOESY spectra, 184 backbone torsion angle restraints, and nine χ-dihedral angle restraints. Representative intermolecular NOE cross-peaks are shown as strip plots in S3 Fig, and the intermolecular NOEs used for the structural calculations are summarized in S4A Fig. Of the 100 independently calculated structures obtained in the final cycle, 20 conformers with the lowest CYANA target function values were further refined by restrained energy minimization and subsequently used for all downstream analyses. The statistics of the structures and distance and torsion angle restraints used in the CYANA program are summarized in Table 1. The root-mean-square deviation (RMSD) from the mean structure was $0.65 \pm 0.20$ Å for the backbone atoms (N, C$^{\alpha}$, C′) and $1.08 \pm 0.17$ Å for all heavy (non-proton) atoms in the well-ordered region, comprising residues 70–118 of UBAh and residues 1–71 of ubiquitin. For

**Table 1. Summary of conformational constraints and structural statistics for 20 energy-refined conformers of the complex between UBAh of HBS1L and ubiquitin.**

| NMR distance and dihedral angle constraints | |
|---|---|
| Distance restraints | |
| Total NOEs | 3,558 |
| Intramolecular NOEs | Total (UBAh, ubiquitin) 3,523 (1,386, 2,137) |
| Sequential ($\lvert i - j \rvert = 1$) | 1,771 (770, 1,001) |
| Medium-range ($1 < \lvert i - j \rvert < 5$) | 702 (356, 346) |
| Long-range ($\lvert i - j \rvert \geq 5$) | 1,050 (260, 790) |
| Intermolecular NOEs between UBAh and ubiquitin | 35 |
| φ/ψ dihedral angle restraints (TALOS) | 184 (81, 103) |
| χ Dihedral angle restraints | 9 (4, 5) |
| **Structure statistics** | |
| AMBER energies (kcal/mol) | |
| Total | −7,109.81 ± 15.70 |
| Distance restraints | 6.74 ± 0.30 |
| Dihedral angle restraints | 1.23 ± 0.20 |
| Ramachandran plot statistics (%) (res. 26-75 (UBAh) and 91-161 (ubiquitin)): | |
| Residues in most favored regions | 93.3; 96.4 |
| Residues in additionally allowed regions | 6.7; 3.6 |
| Residues in generously allowed regions | 0.0; 0.0 |
| Residues in disallowed regions | 0.0; 0.0 |
| Average RMSD from mean coordinates (Å) | |
| Backbone (res. 70-118 and 1-71) | 0.65 ± 0.20 (0.39 ± 0.12, 0.20 ± 0.05) |
| Heavy atoms (res. 70-118 and 1-71) | 1.08 ± 0.17 (1.32 ± 0.18, 0.77 ± 0.08) |

The best-fit superposition of the 20 conformer ensembles is shown in Fig 3A and S4B Fig.

the well-ordered regions of the individual domains, the RMSD values were 0.39±0.12 Å for the backbone and 1.32±0.18 Å for all heavy atoms in UBAh, and 0.20±0.05 Å and 0.77±0.08 Å, respectively, in ubiquitin. Highly disordered regions were observed in the following regions: the N-terminal tag (GSSGSSG), the adjacent segment (Glu51–His67), the C-terminal segment (Asp119–Gly120), and the C-terminal tag (SGPSSG).

For the free form of UBAh, the corresponding data together with the structure are presented in S2 Table and S4C Fig. A comparison of the free and ubiquitin-bound forms of UBAh showed that significant structural changes upon ubiquitin binding occur in the α1/α2 loop and at the C-terminal ends of α1 and α3, as described below (S4D Fig). Hereafter, only the UBAh structure of this complex is shown.

Finally, with regard to the ubiquitin structure bearing the N-terminal seven-residue tag, the NMR results indicate that it adopts the canonical ubiquitin fold, also known as the β-grasp fold, consisting of one α-helix and five β-strands (αA: Ile23–Glu34; β1: Met1–Lys6; β2: Thr12–Val17; β3: Gln41–Phe45; β4: Lys48–Gln49; β5: Thr66–Leu71) (S4E Fig). This structure was virtually identical to that of untagged free ubiquitin. N-terminally tagged ubiquitin has been reported to function properly in structure–function studies [53] and in the determination of complex structures [54]. However, as reported by Nelson et al. [53], slight chemical shift differences between tagged and untagged ubiquitin were confined to a limited region near the N-terminus (Met1), specifically within part of the β-sheet and on one side of αA, suggesting minor structural changes in this region. Because this region lies on the face opposite the Ile44 hydrophobic patch, the seven-residue tag is unlikely to substantially affect the UBAh–ubiquitin interaction mode.

### Structural features of the UBAh domain: conservation across eukaryotes

The NMR results show that the UBAh structure consists of three tightly packed α-helices (α1: 71–88; α2: 94–103; and α3: 108–118), forming a three-helix bundle, with the connecting α1/α2 and α2/α3 loops being well ordered (Fig 3A). Within the UBAh structure, an extensive hydrophobic interaction network forms a large hydrophobic core comprising one residue upstream of α1, six in α1, two in the α1/α2 loop, six in α2, one in the α2/α3 loop, and six in α3, collectively accounting for 44% of the 50 residues in the ordered region (S5A Fig). A portion of this core exposed on the surface contributes substantially to ubiquitin binding, as described below.

The two well-defined loops connecting the α-helices are each characterized by distinct β-turn types. The α1/α2 loop, together with Leu88 in α1, forms a six-residue segment ([88]LGDAVP[93]) that adopts an overlapping β-turn conformation (referred to as a double β-turn). Within the segment, residues [88]LGDA[91] and [90]DAVP[93] correspond to Type II′ and Type VIII β-turns, respectively, according to the β-turn classification [55] (Fig 3B, left; S6A Fig). The type II′ β-turn is canonical, with glycine at the i+1 position, and the main-chain amide of Ala91 (i+3) forms a hydrogen bond with the main-chain carbonyl oxygen of Leu88 (i). The type VIII β-turn represents one of the most common nonclassical β-turn categories. The four-residue segment satisfies the geometric criterion for a β-turn (Cα(i)–Cα(i+3) < 7 Å) and is classified as a type VIII β-turn based on the backbone dihedral angles of the i+1 and i+2 residues. Sequence alignment suggests that residues forming the α1/α2 loop are highly conserved across diverse eukaryotic lineages, including vertebrates, insects, gordiids, and the fission yeast S. pombe, with the exceptions of M. sanguinolenta and S. cerevisiae (see Discussion) (Fig 2B). However, at position 93 (position i+3), the residue can vary (Pro in most species, Met in X. laevis, and Ser in D. melanogaster and S. pombe). Proline is indeed frequently found at the i+3 position in type VIII β-turns, but no strict residue preference is observed at this position [55]. In addition, because the Pro side chain is oriented outward, it does not contribute to the hydrophobic interactions that stabilize the structure. Taken together, these observations suggest that the type VIII β-turn, namely, the double β-turn forming the α1/α2 loop, is largely conserved across eukaryotes, with only a few exceptions, such as these two species. Furthermore, DALI analysis [56], based on the α1–α1/α2 loop–α2 region, indicated that the α1/α2 loop is uniquely characteristic of UBAh domains in both length and architecture among loops connecting two α-helices, including those of the UBA and CUE domains. Accordingly, this loop serves as a hallmark feature distinguishing the UBAh domain from the UBA and CUE domains.

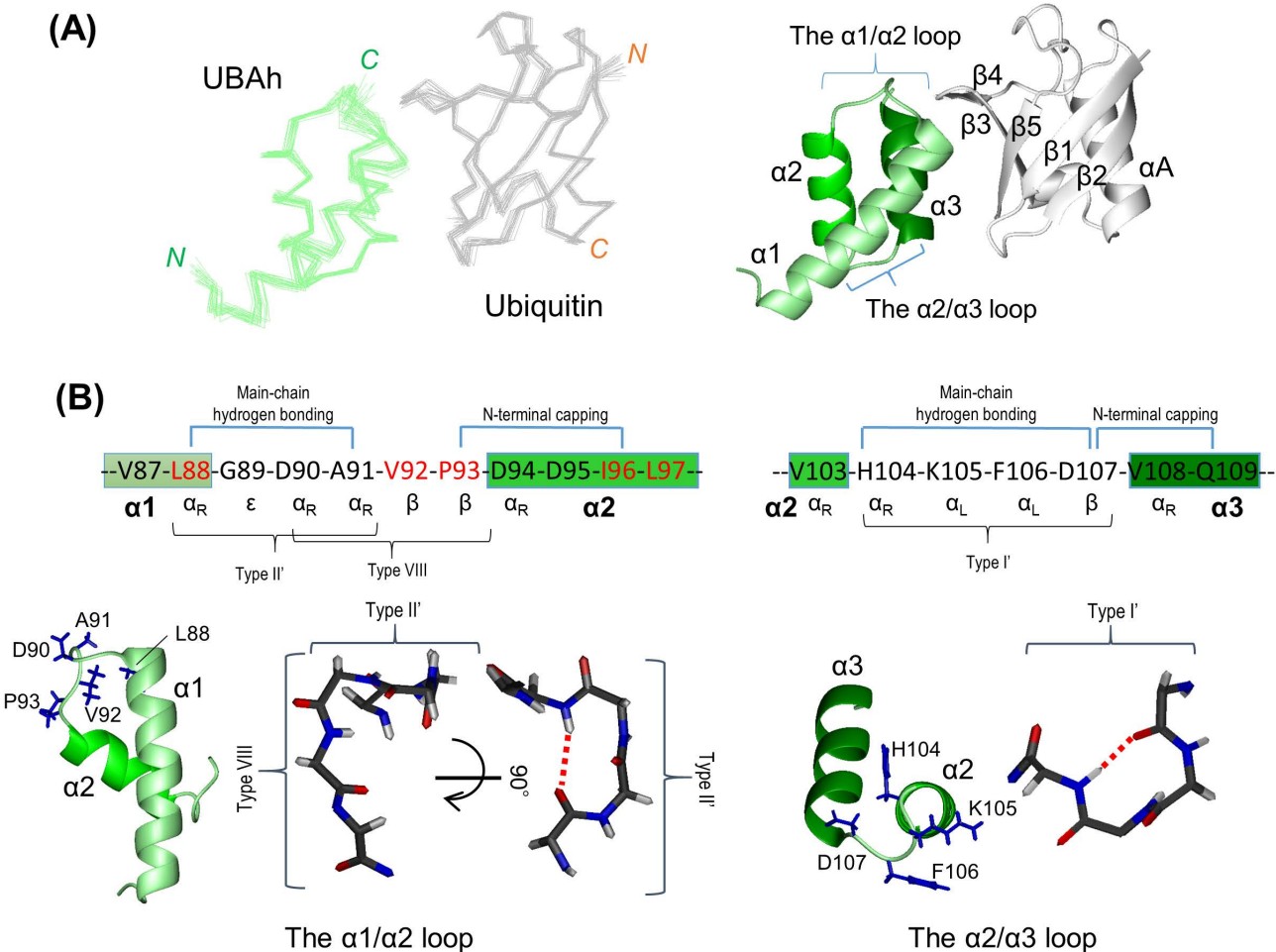

**Fig 3. Structure of the UBAh–ubiquitin complex.** (A) Left: Cᵃ traces of the backbone atoms for the 20 superimposed lowest-energy conformers of the complex. Residues 68–119 of UBAh (green) and residues 1–72 of ubiquitin (gray) are shown. Right: ribbon representation of UBAh in complex with ubiquitin in the same orientation as in the left panel. (B) Characteristic structures of the α1/α2 (left) and α2/α3 (right) loops. In this sequence, the types of β-turns and hydrogen bond connections (blue lines above the sequences) are shown. Side chains of the loop residues are displayed in blue in the ribbon representations. In the stick representations, the red dotted lines indicate hydrogen bonds within the β-turns.

The α2/α3 loop (¹⁰⁴HKFD¹⁰⁷) adopts a type I′ β-turn conformation (Fig 3B, right; S6B Fig). Although type I′ is among the most common β-turns and has no strict residue preference, Phe106 and Asp107 are highly conserved. This is likely because they contribute to the stabilization of the helical bundle. The aromatic ring of Phe106 interacts with the aliphatic portion of Arg76 in α1 and with Leu69 in the upstream region of α1, forming hydrophobic contacts that likely stabilize the interface between the α2/α3 loop and α1 (S5B Fig). Phe/Tyr residues are observed at position 106; Leu/Ile/Thr (methyl group) or Lys (aliphatic side chain) at position 69; and Arg/Lys/Phe/Leu at position 76 (with exceptions such as Cys at position 106 in *D. rerio* and Cys at position 76 in *Gordionus sp.*) (Fig 2B). Thus, similar hydrophobic interactions may be conserved in most eukaryotes. However, because the residues at positions i and i + 1 are not conserved, it remains unclear whether the type I′ β-turn conformation is widely conserved across eukaryotes. Nevertheless, given that the putative turn length was consistently four residues, the overall turn conformation could be maintained. Moreover, the side chain of Asp107 forms a hydrogen bond with the backbone amide and/or the side chain amide of residue 109,

thereby functioning as an N-terminal helix cap of α3 (S5B Fig). As Asp at position 107 is invariant and Gln/Glu/Ala/His are observed at position 109 (Fig 2B), the hydrogen bond with the backbone amide is likely to be more consistently preserved and may play a more central role in helix capping.

## Structural basis of the UBAh–ubiquitin interaction: comparison with UBA and CUE domains

A schematic representation of the main residue–residue interactions is provided in S7 Fig. The complex reveals that the hydrophobic patch centered on Ile44 in β3 of ubiquitin constitutes the primary binding interface with UBAh (Fig 4A and 4B). The hydrophobic patch in ubiquitin is formed by Leu8 (β1/β2 loop), Ile44 (β3), His68 (β5), and Val70 (β5), whereas the corresponding hydrophobic surface in UBAh is composed of Val87 and Leu88 (α1), as well as Leu112 and Leu116 (α3). A similar interaction is observed in UBA and CUE domains [57,58]. The key hydrophobic residues involved in ubiquitin recognition are Val87, Leu112, and Leu116 in UBAh; Met19, Ile43, and Leu47 in CUE; and Met342, Leu365, and Leu369 in UBA (Fig 5A, 5B, and 5C). Although the relative positions of these residues are largely conserved, their overall spatial arrangement is not identical, as the three helices are subtly shifted relative to one another (Fig 5D and 5E). Notably, because the side chain of Val87 in UBAh, the penultimate residue of α1, directly contacts that of Ile44 in β3 of ubiquitin, Val87 appears to be a critical determinant of this hydrophobic interaction. This position is invariably occupied by hydrophobic residues such as Val, Met, or Leu across all members of the THB–Ub group (S8 Fig).

Additionally, a salt bridge and a hydrogen bond may form between α3 of UBAh and β3 of ubiquitin, positioned over the hydrophobic patch–patch interface (Fig 4C). The guanidinium group of Arg42 (β3) in ubiquitin may form a hydrogen bond with the hydroxyl group of Ser113 (α3) and a salt bridge with the carboxylate group of Glu117 (α3), both in UBAh; moreover, it may form an intramolecular hydrogen bond with the amide carbonyl group of Gln49 (β2). Thus, these interactions appear to establish a network centered on Arg42 in ubiquitin. This network likely contributes to the positional shift of α3 relative to α1 and α2 upon ubiquitin binding (S4C and S4D Fig). Notably, even in vertebrates, position 113 is occupied by Ser or Asp, whereas position 117 is occupied by Glu, Ser, Ala, or Lys (Fig 2B). At least one electrostatic interaction is likely to be conserved. Similar electrostatic interactions have been reported in UBA and CUE domains despite differences in position and residue identity [59,60].

Furthermore, two characteristic interactions were observed between the C-terminal region of α1 (Leu88) and the α1/α2 loop of UBAh (89GDAVP93), and the β-turn between β3 and β4 of ubiquitin (45PAGK48): (i) Based on the angle and distance between the C=O and N–H groups, the C=O group of Val87 in UBAh appears to form a water-mediated hydrogen bond with the N–H group of Gly47 in ubiquitin (Fig 4D). This interaction has also been observed in the X-ray ubiquitin complex structures of the UBA domain of the ubiquitin ligase EDD [2QHO] [59] and the CUE-like domain of N4BP1 [8T48] [61] (S9A Fig). Conversely, in the NMR and X-ray ubiquitin complex structures of Dsk2 UBA [1WR1; 4UN2] [62,63], the C=O of the corresponding residue in UBA forms a direct hydrogen bond with the N–H of Gly47 in ubiquitin. Thus, whether a water molecule participates in this interaction appears to vary among the members of the THB–Ub group. This hydrogen bond may be regarded as functioning as an N-terminal capping of α1. (ii) Residues in the α1/α2 loop are involved not only in the interaction but also in maintaining the UBAh structure. Leu88 and the first three residues (89GDA91) of the α1/α2 loop form a Type II′ β-turn, whose conformation orients the side chain of Ala91 to establish a van der Waals contact with the other Hα proton of Gly47 in ubiquitin (S9B Fig). In contrast, the conformation of the other β-turn (Type VIII; 90DAVP93) orients the side chain of the following residue, Val92, inward, enabling hydrophobic interactions with Ile96 in α2 and Val115 in α3 (S9B Fig). Furthermore, the Type II′ β-turn contributes to the formation of a hydrophobic, dish-like cavity (Fig 4B). This cavity accommodates the slight protrusion of the ubiquitin β-turn between β3 and β4; in other words, the Type II′ β-turn is positioned to avoid steric clashes with this ubiquitin β-turn.

## Distinct α1/α2 loop conformations in UBAh, UBA, and CUE domains: functional implications

Although the sequence and/or length of the α1/α2 loop differ among the three domains, the loop adopts distinct conformations that avoid steric clashes with the β3/β4 turn of ubiquitin (Fig 5F; S6A, S6C, S6D and S8 Figs). This loop variation is

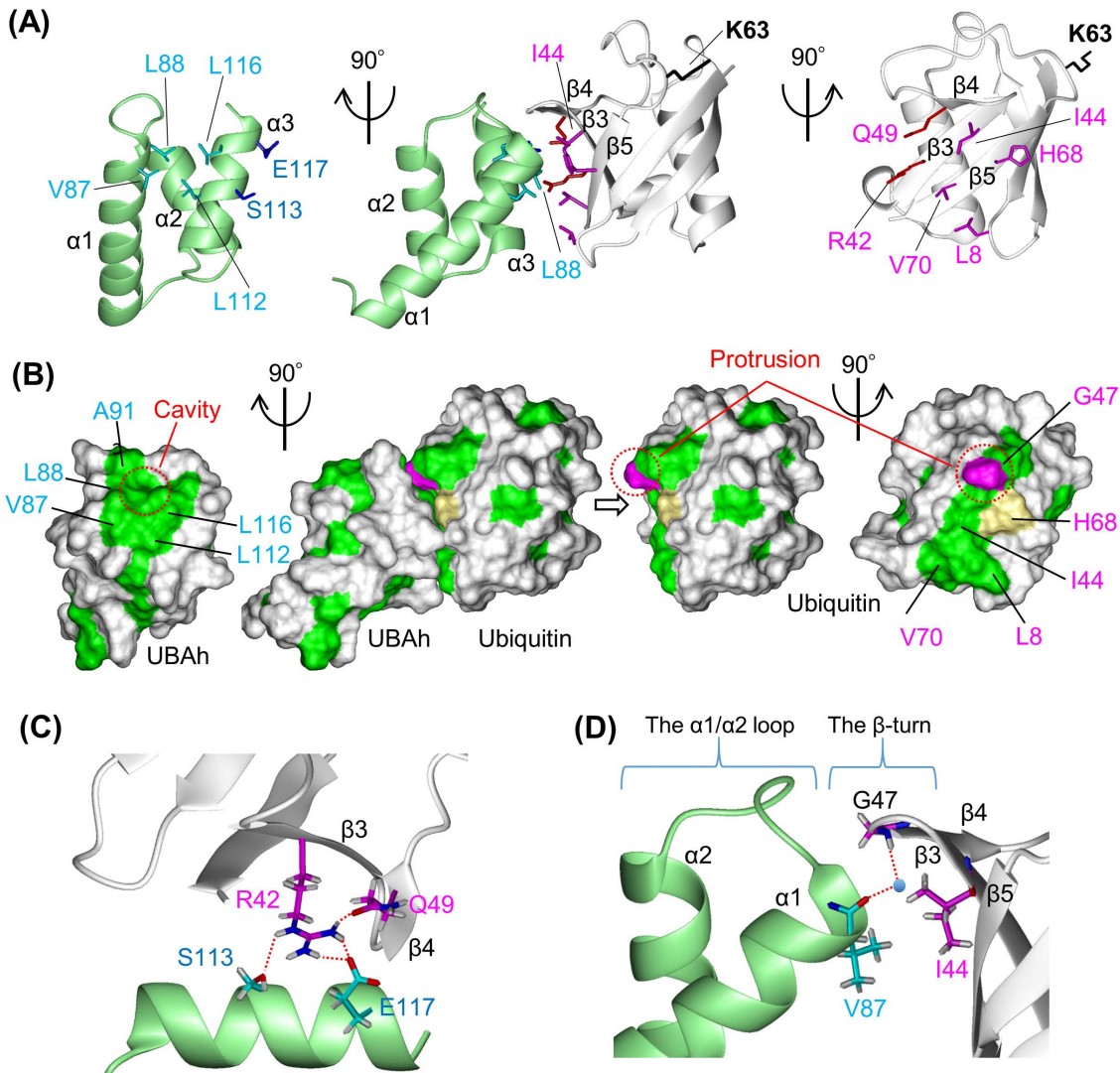

**Fig 4. Interaction of UBAh with ubiquitin.** (A) Ribbon representation of the UBAh–ubiquitin complex. The center panel is shown in the same orientation as Fig 3A. In the left and right panels, UBAh and ubiquitin are shown separately, each in the rotated orientation indicated in the figure. Side chains of residues involved in complex formation are shown, as described in the text. Lys63 in ubiquitin, shown in black, is the linkage residue in ubiquitin chains on the small subunit of stalled ribosomes. (B) Surface representations of the UBAh–ubiquitin complex in the same orientations as in Fig 3A. Hydrophobic residues (Ala, Ile, Leu, Met, Phe, Pro, Trp, and Val) are shown in green, while Gly47 and His68 of ubiquitin are shown in magenta and khaki, respectively. The cavity and protrusion formed mainly by the α1/α2 loop in UBAh and by Gly47 in the ubiquitin β-turn, respectively, are indicated by red dotted circles. (C) Close-up view of the putative interaction network formed by salt bridges and hydrogen bonds between α3 of UBAh and β3 of ubiquitin, with interactions shown as red dotted lines. (D) Close-up view of the putative interaction between the main-chain carbonyl oxygen (C=O) of Val87 in UBAh and the main-chain amide (N–H) of Gly47 in ubiquitin, mediated by a water molecule (blue sphere, position estimated from the UBA crystal structure [PDB ID: 2QHO]). The side chain of Val87 makes direct contact with that of Ile44 in ubiquitin; this hydrophobic contact represents one of the most crucial interactions.

likely to subtly alter the relative positioning of the three α-helices (Fig 5D and 5E), thereby affecting the area and shape of the hydrophobic patch and, consequently, the ubiquitin-binding affinity. These findings further support the conclusion that UBAh is a novel member of the THB–Ub group. Furthermore, sequence alignments among the three domains revealed a distinct hallmark motif in UBAh, located in and downstream of the α1/α2 loop: $^{87}$VLGD/E$^{90}$ (Fig 2B; S8 Fig).

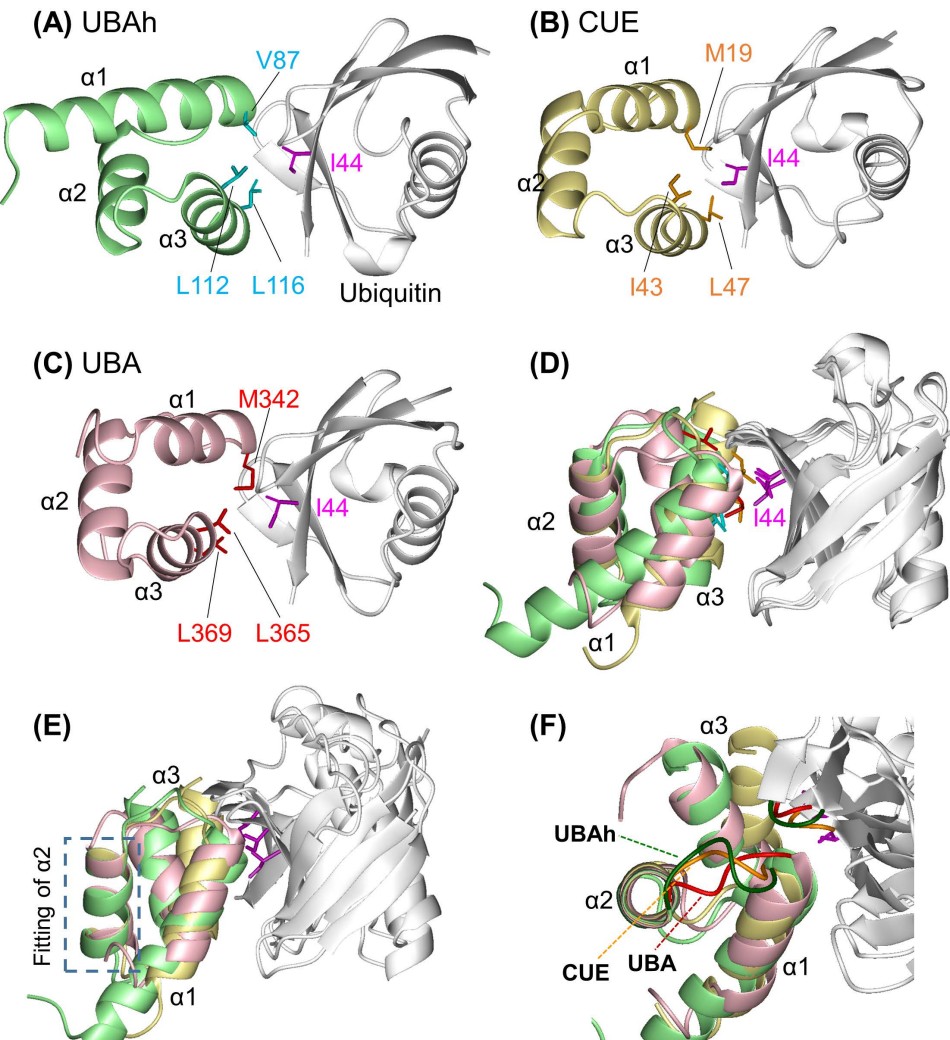

**Fig 5. Structural comparisons among the three-helix bundles in complex with ubiquitin.** (A–C) Ribbon models of ubiquitin (white) complexes with UBAh (pale green), CUE (yellow) [PDB ID: 1OTR], and UBA (pink) [2QHO], shown in the same view as (A). Ile44 in the hydrophobic ubiquitin patch is highlighted in magenta in this panel and in all other related panels. Other key residues involved in ubiquitin binding are shown in various colors. (D) Superimposition of ribbon models of ubiquitin complexes with UBAh, CUE, and UBA aligned on ubiquitin. (E) Superposition of ribbon models of the three complexes, aligned on α2. (F) Close-up view of the α1/α2 loops in the three complexes. The superposition was rotated 50° forward relative to that shown in (E).

The corresponding region contains the [342]MGF[344] motif in UBA and the [19]MFP[21] motif, also known as Hy–F/L–P–X–Hy, in CUE (Fig 5B and 5C; S6 and S8 Figs). These two motifs are hallmarks of UBA and CUE domains, respectively [62,64]. Similar to UBAh, in each domain, these specific residues within the motif also play important roles in ubiquitin recognition and in forming the characteristic structural features of each domain.

### Chemical shift perturbations and affinity determination of the UBAh–ubiquitin complex

We performed chemical shift perturbation analysis using [[15]N-H] heteronuclear single quantum correlation (HSQC) spectroscopy. Chemical shift changes were examined by comparing the spectra of [[15]N]-labeled UBAh alone and in the presence of increasing concentrations of unlabeled ubiquitin (Fig 6; S10 Fig).

**(A)**

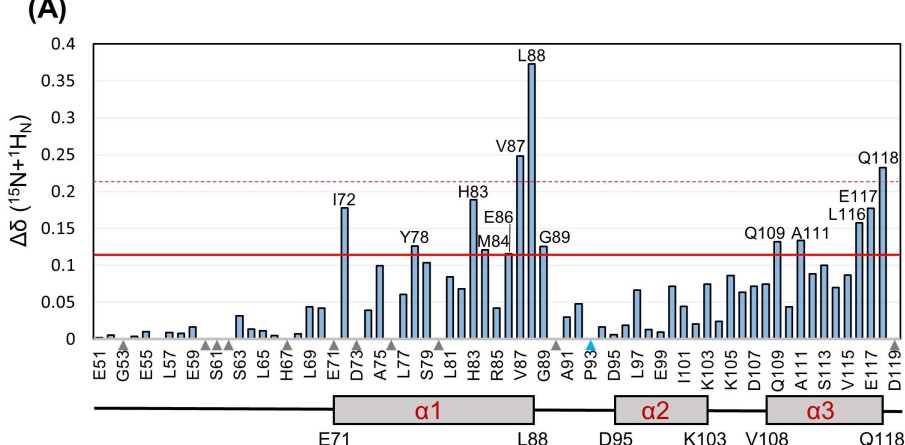

**(B)**

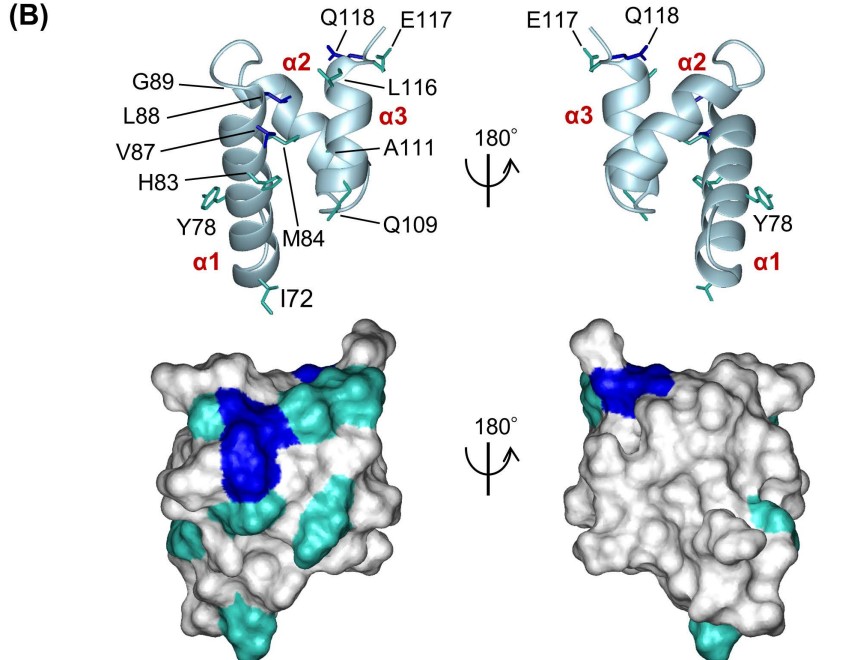

**Fig 6. NMR chemical shift perturbations of labeled UBAh upon binding to non-labeled ubiquitin.** (A) Chemical shift perturbations of labeled UBAh upon ubiquitin binding. Perturbation values were obtained from [$^{1}$H,$^{15}$N]-HSQC spectra in the absence and presence of ubiquitin (S10 Fig). The weighted chemical shift change Δδ ($^{15}$N + $^{1}$H$_N$) for each residue is shown. The red line indicates the average perturbation value (0.0735) plus 0.5 times the standard deviation (0.0711), and the red dotted line indicates the average plus two times the standard deviation. Prolines and residues with resonances that were not assigned after the addition of the peptide are indicated by arrowheads. Only residues with significant chemical shift changes are shown in the graph. (B) Residues with chemical shift changes mapped on the ribbon representation of free UBAh (upper panel) and onto the surface representation (lower panel). The right view was obtained by rotating the left view by 180°. Residues are colored according to the magnitude of the chemical shift change upon ubiquitin binding (blue, above the red dotted line; cyan, between the dotted and solid lines).

First, to assess the structural integrity of the UBAh–ubiquitin complex, we mapped the chemical shift perturbations induced by ubiquitin binding onto the solution structure of UBAh determined in this study using a 1:3 molar ratio (Fig 6). Significant chemical shift changes were observed in residues at the C-terminal ends of α1 and α3, as well as in G89 in the α1/α2 loop, but not in residues in α2. These results are consistent with the binding mode between UBAh and ubiquitin

observed in the complex. Furthermore, as a similar pattern was observed in the Cα RMSD values between the free and ubiquitin-bound UBAh structures (S4C Fig), UBAh likely undergoes subtle conformational changes in α1 and α3 relative to α2 upon ubiquitin binding.

To determine the dissociation constant ($K_d$) for the interaction between UBAh and ubiquitin, we fitted the chemical shift perturbation data to a single-site binding model [52,65]. Global fitting using data from Met84 (α1), Glu86 (α1), and Gly89 (α1/α2 loop) yielded a $K_d$ of 52 μM (S11 Fig). Generally, the $K_d$ for ubiquitin-binding domain–ubiquitin interactions is relatively large, reflecting their weak affinity; UBA domains typically exhibit $K_d$ values of 10–100 μM, whereas CUE domains display weaker affinities of 50–300 μM [58,66,67]. Thus, the measured $K_d$ for the binding of UBAh to ubiquitin is within a reasonable range, suggesting that the complex is moderately stable.

## Discussion

These results demonstrate that the UBAh domain of HBS1L specifically binds to ubiquitin in vitro. Although this binding mode, involving hydrophobic interactions with the Ile44-centered hydrophobic patch of ubiquitin, shares structural features with those of the ubiquitin-binding domains UBA and CUE, substantial differences among the three domains are observed in the sequence and/or length of the α1/α2 loop involved in ubiquitin binding. These two domains are found in various proteins and are often present within the same protein, whereas UBAh is, to our knowledge, exclusively present in HBS1 orthologs. Thus, within the THB–Ub group, UBAh appears to be uniquely specialized for HBS1-mediated recognition of ubiquitin bound to ribosomes. This study also suggests that the classification of THB–Ub group members may be further refined based on the sequence and structural features of loops containing distinct types of β-turns. It would be of interest to examine why HBS1L orthologs employ a single UBAh domain rather than UBA or CUE domains—that is, why different proteins selectively utilize distinct members of the THB–Ub group.

Additionally, the $K_d$ of UBAh for monoubiquitin is approximately 50 μM, which may appear moderate or relatively weak. However, because UBAh is proposed to bind polyubiquitin on the 40S subunit of stalled ribosomes in vivo, the avidity effects arising from polyubiquitin on the ribosomal surface are expected to enhance the effective binding strength by increasing the local ubiquitin concentration and promoting rapid rebinding to adjacent moieties. Indeed, several CUE and UBA domains that bind monoubiquitin have been reported to recognize polyubiquitin with substantially higher affinity [58,68]. For example, the UBA domain of E2-25K, an ubiquitin-conjugating enzyme, binds monoubiquitin with a $K_d$ of ~400 μM, whereas its affinity for polyubiquitin is markedly higher ($K_d$ ~27 μM) [67]. Thus, the moderate affinity for monoubiquitin does not preclude a functional role under physiological conditions.

One conceivable scenario regarding the role of the UBAh–ubiquitin interaction is as follows (Fig 1). As noted in the Introduction, endonucleolytic cleavage during the NGD pathway may generate non-stop ribosomes marked by K63-linked polyubiquitination of specific 40S ribosomal proteins, such as eS10 (RPS10), in humans. This ubiquitination of non-stop ribosomes is likely recognized by the UBAh domain of HBS1L, thereby enhancing the recruitment of the core HBS1L–PELO complex, which contains a GTPase domain, to the vacant A site. Additionally, ubiquitin recognition may help prevent erroneous targeting of the complex to normal ribosomes because ubiquitinated ribosomes are definitive rescue substrates. An important question concerning the ubiquitin-binding domains is whether and how they recognize Lys-linked polyubiquitin chains in vivo. The UBAh-binding mode revealed in this study suggests that UBAh can interact with individual ubiquitin moieties within the chain as K63-linked chains adopt an extended conformation. Nevertheless, the possibility that the K63-linked diubiquitin may engage UBAh, for instance in a "sandwich-like" manner, as observed in certain UBA domains, resulting in an apparent $K_d$ lower than that for a single ubiquitin, cannot be excluded at this stage [66,67]. Thus, further experimental verification is required.

The HBS1L–PELO complex is active on non-stop ribosomes even in the absence of ubiquitination, indicating that the UBAh–ubiquitin interaction is not required. However, the key question is when this interaction is necessary. One possible explanation is that under specific conditions—such as when ubiquitinated stalled ribosomes accumulate due to cellular

stress and the intracellular concentration of the HBS1L–PELO complex becomes significantly lower than that of these ribosomes—the importance of the UBAh–ubiquitin interaction may become more pronounced, facilitating the recognition of aberrant ribosomes. Remarkably, the N-terminal region upstream of UBAh contains at least two Asp/Glu-rich stretches that are well conserved among HBS1L orthologs (S12 Fig). Even in the absence of ubiquitin, it is possible that this region assists in the binding of the HBS1L–PELO complex to ribosomes. Moreover, the N-terminal region contains many Ser/Thr/Tyr residues, and in particular, a Ser/Thr/Tyr-rich region is located between these two stretches (S12 Fig). Phosphorylation of these residues could provide a regulatory platform for signal transduction and the control of subcellular localization [69]. Further in vivo studies will be required to clarify the putative roles of the acidic stretches and Ser/Thr/Tyr-rich region. Their contributions are expected to vary depending on cellular conditions and operate either synergistically or in a complementary manner.

HBS1L, similarly to PELO, is broadly conserved across eukaryotes, and its UBAh domain is also conserved. Nevertheless, our analyses revealed notable exceptions in certain species, including *S. cerevisiae* and *C. elegans,* for which relevant experiments have been described in the Introduction. In *S. cerevisiae* HBS1L, the region corresponding to the UBAh region adopts a three-helix bundle structure (hereafter termed UBAh$^{S.c.}$) However, several residues identified in this study as being critical for ubiquitin interactions differ from those identified in *S. cerevisiae* (Fig 2B). For example, the UBAh-specific motif, $^{87}$VLGDA$^{91}$, is replaced with QLQDY, and Leu116 in α3 is substituted with Lys in UBAh$^{S.c.}$. Furthermore, the superimposition of UBAh$^{S.c.}$ with the ubiquitin-bound UBAh structure suggests that its α1/α2 loop, extended by two residues, sterically clashes with the β-turn of ubiquitin (S13A Fig). Thus, UBAh$^{S.c.}$ is highly unlikely to bind ubiquitin unless it adopts an alternative ubiquitin-binding mode. Interestingly, the cryo-EM structure showed that, while the GTPase domain of HBS1L was bound to the empty A site of the large subunit, UBAh$^{S.c.}$ was bound to a cavity in the 40S subunit, where it interacts with 18S rRNA and the ribosomal protein uS3 (S1C Fig). Unlike many other eukaryotes, this interaction may be substituted with ubiquitin binding [34].

Conversely, in the fission yeast *S. pombe*, sequence alignment indicated that the UBAh of the HBS1L homolog is canonical, namely, of the ubiquitin-binding type (Fig 2B), suggesting divergence among species, even within Ascomycota (S14 Fig). Furthermore, in Basidiomycota, the N-terminal region of HBS1L homologs from *M. sanguinolenta* is presumed to adopt a UBAh fold, based on the conservation of key interaction residues; however, the α1/α2 loop is extended by four residues (Fig 2B). AlphaFold predictions [29] indicate that these four residues form an α-helix, which may help avoid a steric clash with the β-turn of ubiquitin (S13B Fig); thus, the UBAh domain of *M. sanguinolenta* may still bind ubiquitin. Taken together, these observations suggest that fungal UBAh domains may exhibit some degree of structural divergence. Accordingly, the ubiquitin-binding capacity of individual UBAh domains warrants evaluation across fungal species.

Among metazoans, we identified one striking exception: Nematoda. In *C. elegans*, the N-terminal region of Hbs1 shows no sequence homology to UBAh and may not form a structural domain because it is enriched in Pro, Ser, Thr, and Tyr residues (S12 Fig). In contrast, in Nematomorpha, one of the two lineages of Nematoida, the Hbs1 ortholog from *Gordionus montsenyensis sp. nov.* (Montseny hairworm) appears to adopt a canonical UBAh (S14 Fig). In other words, within Ecdysozoa, Hbs1 orthologs in lineages other than Nematoda are thought to retain UBAh. The reason why UBAh is lost in Hbs1 in Nematoda, but is retained in Nematomorpha, remains unknown.

Taken together, extensive sequence alignment indicates that in opisthokonts (i.e., animals and fungi), UBAh in HBS1L is missing in only a few species and is widely conserved in the majority (S14 Fig). However, in plants, the N-terminal region of HBS1L homologs reportedly contains a C4-type zinc-finger domain instead of UBAh, which is another type of ubiquitin-binding domain [28,70]. Therefore, in the early stages of eukaryotic evolution, ubiquitination of the small subunit of stalled ribosomes may have served as a recognition signal for HBS1L in many species through a ubiquitin-binding domain, the specific type of which differs between opisthokonts and plants. In contrast, because the UBAh domain of *S. cerevisiae* can bind directly to the small ribosomal subunit, there may have been an evolutionary stage in

which ubiquitin-dependent and -independent modes of UBAh binding coexisted, functioning either mutually exclusively or sequentially. Such dual functionality may still be retained in some extant species. Further studies across diverse species are required to confirm this hypothesis.

Notably, UBAh, together with acidic stretches, is also present in the N-terminal region of SKI7, an isoform of the *HBS1L* gene; the remaining region is specific to SKI7 and serves as a platform for complex interactions [71]. Although SKI7 and HBS1 are highly conserved in vertebrates, SKI7 homologs in other species frequently show substantial sequence divergence and are not always encoded by the *HBS1L* gene (e.g., *S. cerevisiae* Ski7 lacks UBAh^sc). SKI7 serves as an essential bridge between the SKI complex, which possesses RNA helicase activity, and the RNA exosome, which contains the $3' \rightarrow 5'$ exonuclease (Fig 1). A recent cryo-EM analysis in humans demonstrated that in the NGD pathway, SKI7 recruits exosomes (EXO10) to the SKI complex associated with the 40S subunit of ubiquitinated stalled ribosomes [72]. Furthermore, the N-terminal fragment of SKI7 containing UBAh has been shown to associate with the 40S subunit; however, the precise binding site within the ribosome remains undefined. Therefore, elucidation of the direct binding partners of UBAh is of considerable interest. Should it prove to be ubiquitin, such an interaction could promote the recruitment of the SKI complex to stalled ribosomes and strengthen its ribosomal affinity, a mechanism that may be analogous to the mechanism proposed for the UBAh of HBS1L.

Finally, by revealing that HBS1L and SKI7, similar to several other factors involved in ribosome rescue, can recognize ubiquitin, this study helps fill a gap in our current understanding (Fig 1). These structural insights, together with the extensive sequence alignments presented in this study, are expected to facilitate future experimental validation of UBAh–ubiquitin interactions of HBS1L and SKI7 beyond *S. cerevisiae* and *C. elegans*.

## Supporting information

**S1 Table. Completeness of resonance assignments in the UBAh domain and its complex with Ubiqusitin at pH 6.0 and 298 K.**
(PDF)

**S2 Table. Summary of conformational constraints and structural statistics for 20 energy-refined conformers of UBAh in its free form.**
(PDF)

**S1 Fig. Structure of eukaryotic 80S ribosomes.**
(PDF)

**S2 Fig. Assigned 2D ^1H-^15N HSQC spectrum of the UBAh domain in the presence of ubiquitin (1:1 molar ratio).**
(PDF)

**S3 Fig. Representative intermolecular NOE strip plots illustrating contacts between UBAh and ubiquitin (2D strips from a 3D ^13C-edited NOESY spectrum).**
(PDF)

**S4 Fig. Solution structure of the UBAh–ubiquitin complex and comparison of UBAh in its free and bound forms.**
(PDF)

**S5 Fig. Structural features of UBAh.**
(PDF)

**S6 Fig. Ramachandran plots of residues forming the α1/α2 (A) and α2/α3 loop (B) in UBAh, the α1/α2 loop (C) in UBA [1WR1], and the α1/α2 loop (D) in CUE [1OTR].**
(PDF)

 

**S7 Fig. Schematic diagram of the interactions between UBAh and ubiquitin.**
(PDF)

**S8 Fig. Amino acid sequence alignments of UBA, CUE, and UBAh domains.**
(PDF)

**S9 Fig. Characteristic interaction sites between UBAh and ubiquitin.**
(PDF)

**S10 Fig. [¹H, ¹⁵N]-HSQC spectra of labeled UBAh at different ubiquitin concentrations.**
(PDF)

**S11 Fig. Chemical shift perturbation titration curves for the interaction between ¹⁵N-labeled UBAh and unlabeled ubiquitin.**
(PDF)

**S12 Fig. Amino acid alignment of the N-terminal region upstream of the UBAh domain of HBS1L/SKI7 from various eukaryotes.**
(PDF)

**S13 Fig. Structural comparison of mouse UBAh (from the UBAh–ubiquitin complex) between other UBAh structures.**
(PDF)

**S14 Fig. Simplified phylogenetic tree of major eukaryotes indicating the presence of UBAh in HBS1L.**
(PDF)

**S1 File. Raw data for Fig 6A, S4D Fig, and S11 Fig.**
(XLSX)

## Acknowledgments

We are deeply grateful to the RIKEN Structural Genomics/Proteomics Initiative (RSGI) and the National Project on Protein Structural and Functional Analyses of the Ministry of Education, Culture, Sports, Science, and Technology of Japan for providing the initial opportunity to begin this structural study.

## Author contributions

**Conceptualization:** Kanako Kuwasako, Nobukazu Nameki, Yutaka Muto.

**Formal analysis:** Kanako Kuwasako, Nobukazu Nameki, Minako Okada.

**Funding acquisition:** Kanako Kuwasako, Yutaka Muto.

**Investigation:** Fahu He, Mari Takahashi, Kengo Tsuda, Takashi Nagata, Peter Güntert, Naohiro Kobayashi, Takanori Kigawa, Mikako Shirouzu, Akiko Tanaka, Shigeyuki Yokoyama, Yutaka Muto.

**Supervision:** Kanako Kuwasako.

**Visualization:** Kanako Kuwasako, Nobukazu Nameki.

**Writing – original draft:** Nobukazu Nameki.

**Writing – review & editing:** Kanako Kuwasako, Fahu He, Minako Okada, Mari Takahashi, Kengo Tsuda, Takashi Nagata, Peter Güntert, Naohiro Kobayashi, Takanori Kigawa, Mikako Shirouzu, Akiko Tanaka, Shigeyuki Yokoyama, Yutaka Muto.

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
