## [Decision Letter · Decision Letter 0]

9 Feb 2026

PONE-D-25-67488Solution structure of mouse HBS1L/SKI7-specific UBA domain in complex with ubiquitin: Implications for stalled ribosome recognitionPLOS One

Dear Dr. Kuwasako,

Thank you for submitting your manuscript to PLOS ONE. After careful consideration, we feel that it has merit but does not fully meet PLOS ONE’s publication criteria as it currently stands. Therefore, we invite you to submit a revised version of the manuscript that addresses the points raised during the review process.

We look forward to receiving your revised manuscript.

Kind regards,

Shailender Kumar Verma, Ph.D.

Academic Editor

PLOS One

Journal Requirements:

2. We note that this submission includes NMR spectroscopy data. We would recommend that you include the following information in your methods section or as Supporting Information files:

(1) The make/source of the NMR instrument used in your study, as well as the magnetic field strength. For each individual experiment, please also list: the nucleus being measured; the sample concentration; the solvent in which the sample is dissolved and if solvent signal suppression was used; the reference standard and the temperature.

(2) A list of the chemical shifts for all compounds characterised by NMR spectroscopy, specifying, where relevant: the chemical shift (δ), the multiplicity and the coupling constants (in Hz), for the appropriate nuclei used for assignment.

(3)The full integrated NMR spectrum, clearly labelled with the compound name and chemical structure.

We also strongly encourage authors to provide primary NMR data files, in particular for new compounds which have not been characterised in the existing literature. Authors should provide the acquisition data, FID files and processing parameters for each experiment, clearly labelled with the compound name and identifier, as well as a structure file for each provided dataset. See our list of recommended repositories here: https://journals.plos.org/plosone/s/recommended-repositories

“This work was supported by JSPS KAKENHI (grant number 22K06105, to K.K.) and a Daigaku Tokubetsu Kenkyuhi grant from Musashino University (to Y.M. and K.K.).”

Reviewers' comments:

Reviewer's Responses to Questions

**Comments to the Author**

1. Is the manuscript technically sound, and do the data support the conclusions?

Reviewer #1: Yes

Reviewer #2: Yes

2. Has the statistical analysis been performed appropriately and rigorously? 

Reviewer #1: Yes

Reviewer #2: N/A

3. Have the authors made all data underlying the findings in their manuscript fully available?

Reviewer #1: Yes

Reviewer #2: Yes

4. Is the manuscript presented in an intelligible fashion and written in standard English?

Reviewer #1: Yes

Reviewer #2: Yes

5. Review Comments to the Author

Reviewer #1: In this manuscript, the authors structurally characterized a murine HBS1L/SK17-specific UBA domain complexed with ubiquitin using solution NMR spectroscopy. This manuscript presents new structural data of the ubiquitin-binding domain (termed UBAh) of HBS1L/SK17, thereby providing valuable insights into the molecular mechanism mediated by ubiquitin-binding proteins. This reviewer would recommend this manuscript for publication in PLoS ONE. However, there are several concerns that should be adequately addressed during the manuscript revision.

Comments:

1. Comparing the results presented in the manuscript, the introduction seems overly detailed, which makes the main points difficult to grasp. Specifically, the section from page 7, line 8 (from the bottom), to page 9, line 2 (from the top), is appropriate. However, the preceding section, spanning approximately four pages, is quite lengthy. It would be helpful if the authors could refine this section to be more concise and focused.

2. The authors determined the solution structures of UBAh in its free form and in complex with ubiquitin, and subsequently conducted a structural comparison. The authors concluded that ubiquitin binding induces negligible structural changes in UBAh. However, as shown in Figure S4C, the alpha3 helix involved in ubiquitin interaction appeared to be significantly tilted, in contrast to the alpha1 and alpha2 helices. Although the overall RMSD value is reported as 0.825A, presenting a graph showing the RMSD for each individual amino acid residue would allow for a more objective assessment.

3. The authors used a modified ubiquitin with extra 7AA N-terminal tag for the NMR spectroscopy experiments. To this reviewer's knowledge, most of the (mono)ubiquitin structural biology studies have used non-tagged ubiquitin. The N-termini is close to K63 and is also a modification site for linear Ub chains. Therefore, the authors should show a functional integrity of the modified Ub as compared with the authentic Ub. For example, this could be done by performing a chemical shift perturbation experiment using a commercially available ubiquitin, or by introducing reference example(s) utilizing a modified mono-ubiquitin with an N-terminal tag.

Reviewer #2: In this manuscript, Nameki and He et al. presents a high-quality solution NMR structures of the apo and ubiquitin-bound mouse HBS1L/SKI7-specific UBA domain. The NMR data are rigorous and well-determined , and the interface analysis is convincing. The conclusions are largely appropriate, provided that functional implications are framed with appropriate caution.

Overall, this work provides a valuable structural basis for understanding how HBS1L/SKI7 may engage ubiquitin during stalled ribosome surveillance. However, the conceptual framing of UBAh as a "new member" of the THB–Ub group requires more evidence or discussion, and the functional interpretations should be more cautiously phrased before publication.

Major comments:

1.

The authors define UBAh as a distinct member of the THB–Ub group based on a specific analysis of the turn types in the α1/α2 and α2/α3 loops. But, this classification needs more cautious discussion:

The authors noted that the turn configurations define this specific type. However, several residues in these loops—such as Phe93, His104, and Lys105 (Fig. 2B)—are not highly conserved across species. These variations could alter the turn geometry in non-mammalian homologs. The authors should provide a more detailed comparative structural analysis (eg. via https://consurf.tau.ac.il/) to support the claim that these turn types are a defining feature of "UBAh" across its entire evolutionary range. Otherwise, the claim should be restricted to mammalian UBAh.

On the other hand, if such specific turn-type criteria are used to define a new domain class, could existing UBA or CUE domains also be subdivided into more specific categories (e.g., "UBAc" for CUE)? The authors should discuss whether UBAh is uniquely specialized or if they are proposing a more granular classification system for the entire THB–Ub group. Last, In Fig. S9, the authors compare UBA and CUE domains between yeast and humans. It would be beneficial to also include alignments showing the inter-species conservation of UBA and Cue.

2.

The structural data demonstrate that UBAh binds ubiquitin in vitro, but the functional role remains hypothetical. The manuscript should more explicitly acknowledge that the proposed role of UBAh in stalled ribosome recognition is based on structural compatibility rather than direct in vivo functional validation.

3.

The reported Kd of ∼50 μM is relatively weak compared to many high-affinity ubiquitin receptors. Given the high competition for ubiquitin in the cellular environment, the authors should discuss whether this affinity is sufficient for physiological function. Could avidity effects (via multi-valent interactions) or local concentration increases at the stalled ribosome compensate for this modest affinity?

Minors

1. The evolutionary comparison across species (e.g., yeast, nematodes, and metazoans) is interesting but largely speculative. It would be helpful to more explicitly label these interpretations as hypotheses rather than established conclusions.

2. The discussion comparing direct ribosome binding in yeast versus ubiquitin-mediated recruitment in metazoans could be expanded slightly to address whether these modes might be mutually exclusive or potentially sequential.

3. The section discussing acidic stretches and possible phosphorylation sites is intriguing but speculative. Minor rephrasing to emphasize these as future directions would improve balance.

4. Figure 1 is information-dense. Improving visual hierarchy or providing clearer panel descriptions would help the reader.

5. Briefly clarify if global fitting was used for titration data or how residue-specific deviations were handled.

6. (Page 14, Para 4): The phrase "These two ordered loops" is somewhat contradictory, as "ordered" usually implies rigidity, while loops are inherently flexible.

Probably "two well-defined loops" or "two structurally constrained loops" is better.

6. PLOS authors have the option to publish the peer review history of their article (what does this mean?). If published, this will include your full peer review and any attached files.

Reviewer #1: No

Reviewer #2: No

---

## [Author Response · Author response to Decision Letter 1]

7 Apr 2026

Journal Requirements:

We confirm that our manuscript meets the PLOS One style requirements, including those regarding file naming.

2. We note that this submission includes NMR spectroscopy data. We would recommend that you include the following information in your methods section or as Supporting Information files:

(1) The make/source of the NMR instrument used in your study, as well as the magnetic field strength. For each individual experiment, please also list: the nucleus being measured; the sample concentration; the solvent in which the sample is dissolved and if solvent signal suppression was used; the reference standard and the temperature.

The information has already been described in the previous version. In this revised version, we have explicitly specified the instruments used to improve clarity.

(2) A list of the chemical shifts for all compounds characterised by NMR spectroscopy, specifying, where relevant: the chemical shift (δ), the multiplicity and the coupling constants (in Hz), for the appropriate nuclei used for assignment.

The chemical shift assignments of the amino acid residues (¹H, ¹⁵N, and ¹³C) have been deposited in the Biological Magnetic Resonance Data Bank (BMRB) (accession numbers 36786 and 36787), where the complete chemical shift list will be available after publication. The completeness of the assignments is summarized in S1 Table, and the BMRB accession numbers are provided in the Materials and Methods section.

(3)The full integrated NMR spectrum, clearly labelled with the compound name and chemical structure.

We also strongly encourage authors to provide primary NMR data files, in particular for new compounds which have not been characterised in the existing literature. Authors should provide the acquisition data, FID files and processing parameters for each experiment, clearly labelled with the compound name and identifier, as well as a structure file for each provided dataset. See our list of recommended repositories here: https://journals.plos.org/plosone/s/recommended-repositories

This paper presents a typical NMR structure determination study using ¹⁵N- and/or ¹³C-labeled proteins composed of the 20 standard amino acids. Therefore, novel chemical compounds were not investigated in this study. NMR chemical shift assignments for the protein were deposited in the Biological Magnetic Resonance Data Bank (BMRB) (accession numbers 36786 and 36787), and the NOE lists used for structure calculations were also deposited in the Protein Data Bank (PDB) (accession numbers 9WPR and 9WPS).

Moreover, we have added the full 2D ¹H–¹⁵N HSQC spectrum of the UBAh domain in the presence of ubiquitin (1:1 molar ratio), with peaks labeled with the corresponding amino acid residues (S2 Fig), as well as representative intermolecular NOE strip plots illustrating contacts between UBAh and ubiquitin (2D strips from a 3D ¹³C-edited NOESY spectrum) (S3 Fig).

“This work was supported by JSPS KAKENHI (grant number 22K06105, to K.K.) and a Daigaku Tokubetsu Kenkyuhi grant from Musashino University (to Y.M. and K.K.).”

Please include this amended Role of Funder statement in your cover letter; we will change the online submission form on your behalf

As requested, we have included the following statement in the cover letter: ‘The funders had no role in the study design, data collection and analysis, decision to publish, or preparation of the manuscript.’

4. When completing the data availability statement of the submission form, you indicated that you will make your data available on acceptance. We strongly recommend all authors decide on a data sharing plan before acceptance, as the process can be lengthy and hold up publication timelines. Please note that, though access restrictions are acceptable now, your entire data will need to be made freely accessible if your manuscript is accepted for publication. This policy applies to all data except where public deposition would breach compliance with the protocol approved by the research ethics board. If you are unable to adhere to our open data policy, please kindly revise your statement to explain your reasoning and we will seek the editor's input on an exemption. Please be assured that, once you have provided your new statement, the assessment of your exemption will not hold up the peer review process.

We agree with the journal’s open data policy and will make all relevant data freely available upon publication.

The reviewers’ comments did not include any recommendation to cite specific previously published works.

5. Review Comments to the Author

First, we thank the reviewers for their insightful and constructive comments, which have greatly improved the quality of our manuscript.

We apologize for the oversight regarding the β-turn type of the α1/α2 loop (previously described as type IVb; now corrected to type VIII). We have corrected this in the revised manuscript.

For readability, we have divided the subsection “Structural basis of the UBAh–ubiquitin interaction” in the Results section into two subsections: “Structural basis of the UBAh–ubiquitin interaction: comparison with UBA and CUE domains” and “Distinct α1/α2 loop conformations in UBAh, UBA, and CUE domains: functional implications”, without altering the content.

Reviewer #1: In this manuscript, the authors structurally characterized a murine HBS1L/SK17-specific UBA domain complexed with ubiquitin using solution NMR spectroscopy. This manuscript presents new structural data of the ubiquitin-binding domain (termed UBAh) of HBS1L/SK17, thereby providing valuable insights into the molecular mechanism mediated by ubiquitin-binding proteins. This reviewer would recommend this manuscript for publication in PLoS ONE. However, there are several concerns that should be adequately addressed during the manuscript revision.

Comments:

1. Comparing the results presented in the manuscript, the introduction seems overly detailed, which makes the main points difficult to grasp. Specifically, the section from page 7, line 8 (from the bottom), to page 9, line 2 (from the top), is appropriate. However, the preceding section, spanning approximately four pages, is quite lengthy. It would be helpful if the authors could refine this section to be more concise and focused.

The length appears to be due to the inclusion of long legends for Figs 1 and 2, in accordance with the PLOS One guidelines. As mentioned, the Introduction is still lengthy. Therefore, we have shortened the preceding section from 970 to 666 words (the total length of the Introduction is now within three pages, excluding legends). In addition, Fig 1 has been revised to improve clarity.

2. The authors determined the solution structures of UBAh in its free form and in complex with ubiquitin and subsequently conducted a structural comparison. The authors concluded that ubiquitin binding induces negligible structural changes in UBAh. However, as shown in Figure S4C, the alpha3 helix involved in ubiquitin interaction appeared to be significantly tilted, in contrast to the alpha1 and alpha2 helices. Although the overall RMSD value is reported as 0.825A, presenting a graph showing the RMSD for each individual amino acid residue would allow for a more objective assessment.

Thank you for your valuable suggestion. Our analysis indicates that α3 is significantly tilted upon ubiquitin binding. We have added a new graph showing the per-residue Cα RMSD (S4E Fig) and included the following sentences in the manuscript:

Page 13, line 22: A comparison of the free and ubiquitin-bound forms of UBAh showed that significant structural changes upon ubiquitin binding occur in the α1/α2 loop and at the C-terminal ends of α1 and α3, as described below (S4D Fig).

Page 17, line 12: This network likely contributes to the positional shift of α3 relative to α1 and α2 upon ubiquitin binding (S4C and S4D Fig)

Page 19, line 17: Furthermore, as a similar pattern was observed in the Cα RMSD values between the free and ubiquitin-bound UBAh structures (S4C Fig), UBAh likely undergoes subtle conformational changes in α1 and α3 relative to α2 upon ubiquitin binding.

3. The authors used a modified ubiquitin with extra 7AA N-terminal tag for the NMR spectroscopy experiments. To this reviewer's knowledge, most of the (mono)ubiquitin structural biology studies have used non-tagged ubiquitin. The N-termini is close to K63 and is also a modification site for linear Ub chains. Therefore, the authors should show a functional integrity of the modified Ub as compared with the authentic Ub. For example, this could be done by performing a chemical shift perturbation experiment using a commercially available ubiquitin, or by introducing reference example(s) utilizing a modified mono-ubiquitin with an N-terminal tag.

Nelson et al. performed an NMR-based structural comparison of untagged ubiquitin (residues 1–76, wild type) and its variants containing either a Strep tag plus (GS)₂ or a (GS)₂ tag [9]. In that study, backbone residual dipolar coupling (RDC) analyses of ubiquitin fused to either a Strep tag or (GS)₂ demonstrated that the tagged ubiquitin adopts the same fold in solution as wild-type ubiquitin. Additionally, comparison of chemical shifts between wild-type ubiquitin and (GS)₂-tagged ubiquitin revealed perturbations primarily at Q2 (Δδ_H/N ≈ 0.75 ppm), S57 (≈ 0.13 ppm), and E18 (≈ 0.25 ppm), and to a lesser extent at K63 (≈ 0.05 ppm). The weighted chemical shift difference was calculated as Δδ_H/N = { (Δδ_HN)² + (0.154 × Δδ_N)² }^1/2.

Together, the affected residues clustered within part of the β-sheet and on one side of the α1 helix, centered around the N-terminus (Met1) of ubiquitin, suggesting minor structural changes in this region. Since this region is located on the diametrically opposite face of the protein from the Ile44 hydrophobic patch, the presence of a 7-residue tag is unlikely to substantially alter the interaction mode of UBAh with ubiquitin.

Thus, we have added the following sentences:

Page 13, line 26: Finally, with regard to the ubiquitin structure bearing the N-terminal seven-residue tag, the NMR results indicate…….

Page 13, line 30: N-terminally tagged ubiquitin has been reported to function properly in structure–function studies [52] and in the determination of complex structures [53]. However, as reported by Nelson et al. [52], slight chemical shift differences between tagged and untagged ubiquitin were confined to a limited region near the N-terminus (Met1), specifically within part of the β-sheet and on one side of αA, suggesting minor structural changes in this region. Because this region lies on the face opposite the Ile44 hydrophobic patch, the seven-residue tag is unlikely to substantially affect the UBAh–ubiquitin interaction mode.

52. Nelson SL, Li Y, Chen Y, Deshmukh L. Avidity-Based Method for the Efficient Generation of Monoubiquitinated Recombinant Proteins. J Am Chem Soc. 2023; 145(14): 7748-7752. https://doi.org/10.1021/jacs.3c01943. PMID: 37010382.

53. Lv Z, Rickman KA, Yuan L, Williams K, Selvam SP, Woosley AN, et al. S. pombe Uba1-Ubc15 Structure Reveals a Novel Regulatory Mechanism of Ubiquitin E2 Activity. Mol Cell. 2017; 65(4): 699-714.e696. https://doi.org/10.1016/j.molcel.2017.01.008. PMID: 28162934.

Reviewer #2: In this manuscript, Nameki and He et al. presents a high-quality solution NMR structures of the apo and ubiquitin-bound mouse HBS1L/SKI7-specific UBA domain. The NMR data are rigorous and well-determined, and the interface analysis is convincing. The conclusions are largely appropriate, provided that functional implications are framed with appropriate caution.

Overall, this work provides a valuable structural basis for understanding how HBS1L/SKI7 may engage ubiquitin during stalled ribosome surveillance. However, the conceptual framing of UBAh as a "new member" of the THB–Ub group requires more evidence or discussion, and the functional interpretations should be more cautiously phrased before publication.

Major comments:

1.

The authors define UBAh as a distinct member of the THB–Ub group based on a specific analysis of the turn types in the α1/α2 and α2/α3 loops. But, this classification needs more cautious discussion:

The authors noted that the turn configurations define this specific type. However, several residues in these loops—such as Phe93, His104, and Lys105 (Fig. 2B)—are not highly conserved across species. These variations could alter the turn geometry in non-mammalian homologs. The authors should provide a more detailed comparative structural analysis (eg. via https://consurf.tau.ac.il/) to support the claim that these turn types are a defining feature of "UBAh" across its entire evolutionary range. Otherwise, the claim should be restricted to mammalian UBAh.

As suggested, we revised the manuscript to provide a more detailed description of the loops and their conservation among species (page 14, lines 19, 27; page 15, line 11).

Unfortunately, the recommended website was inaccessible. Instead, we used the DALI server for structural comparison and confirmed that the structure of the α1 helix–α1/α2 loop–α2 helix element, including the sequence, is unique to UBAh domains. Additionally, we have clarified that it remains unclear whether the turn geometry of the α2/α3 loop is conserved across eukaryotes.

Page 15, line 1: Furthermore, DALI analysis [55], based on the α1–α1/α2 loop–α2 region, indicated that the α1/α2 loop is uniquely characteristic of UBAh domains in both length and architecture among loops connecting two α-helices, including those of the UBA and CUE domains. Accordingly, this loop serves as a hallmark feature distinguishing the UBAh domain from the UBA and CUE domains.

Page 15, line 15: However, because the residues at positions i and i+1 are not conserved, it remains unclear whether the type I′ β-turn conformation is widely conserved across eukaryotes. Nevertheless, given that the putative turn length was consistently four residues, the overall turn conformation could be maintained.

On the other hand, if such specific turn-type criteria are used to define a new domain class, could existing UBA or CUE domains also be subdivided into more specific categories (e.g., "UBAc" for CUE)? The authors should discuss whether UBAh is uniquely specialized or if they are proposing a more granular classification system for the entire THB–Ub group. Last, In Fig. S9, the authors compare UBA and CUE domains between yeast and humans. It would be beneficial to also include alignments showing the inter-species conservation of UBA and Cue.

As pointed out, we agree that existing UBA and CUE domains could potentially be further subdivided into more specific categories based on β-turn types. How

---

## [Decision Letter · Decision Letter 1]

23 Apr 2026

Solution structure of mouse HBS1L/SKI7-specific UBA domain in complex with ubiquitin: Implications for stalled ribosome recognition

PONE-D-25-67488R1

Dear Dr. Kuwasako,

We’re pleased to inform you that your manuscript has been judged scientifically suitable for publication and will be formally accepted for publication once it meets all outstanding technical requirements.

Kind regards,

Shailender Kumar Verma, Ph.D.

Academic Editor

PLOS One

Additional Editor Comments (optional):

Reviewers' comments:

Reviewer's Responses to Questions

**Comments to the Author**

1. If the authors have adequately addressed your comments raised in a previous round of review and you feel that this manuscript is now acceptable for publication, you may indicate that here to bypass the “Comments to the Author” section, enter your conflict of interest statement in the “Confidential to Editor” section, and submit your "Accept" recommendation.

Reviewer #1: All comments have been addressed

Reviewer #2: (No Response)

2. Is the manuscript technically sound, and do the data support the conclusions?

Reviewer #1: Yes

Reviewer #2: Yes

3. Has the statistical analysis been performed appropriately and rigorously? 

Reviewer #1: N/A

Reviewer #2: Yes

4. Have the authors made all data underlying the findings in their manuscript fully available?

Reviewer #1: Yes

Reviewer #2: Yes

5. Is the manuscript presented in an intelligible fashion and written in standard English?

Reviewer #1: Yes

Reviewer #2: Yes

6. Review Comments to the Author

Reviewer #1: The authors have adequately addressed all of this reviewer’s comments in the revised manuscript. This reviewer would recommend this manuscript for publication in PLoS ONE.

Reviewer #2: The author made an excellent revision, which well addressed my previous major and minor concerns. I believe this revised version is suitable for publication in PLOS One now.

7. PLOS authors have the option to publish the peer review history of their article (what does this mean?). If published, this will include your full peer review and any attached files.

Reviewer #1: No

Reviewer #2: No

---

## [Editor Report · Acceptance letter]

PONE-D-25-67488R1

PLOS One

Dear Dr. Kuwasako,

I'm pleased to inform you that your manuscript has been deemed suitable for publication in PLOS One. Congratulations! Your manuscript is now being handed over to our production team.

Kind regards,

on behalf of

Dr. Shailender Kumar Verma

Academic Editor

PLOS One